# Clinical Characteristics of Human Mpox (Monkeypox) in 2022: A Systematic Review and Meta-Analysis

**DOI:** 10.3390/pathogens12010146

**Published:** 2023-01-15

**Authors:** Qi Liu, Leiwen Fu, Bingyi Wang, Yinghui Sun, Xinsheng Wu, Xin Peng, Yuwei Li, Yi-Fan Lin, Thomas Fitzpatrick, Sten H. Vermund, Huachun Zou

**Affiliations:** 1School of Public Health (Shenzhen), Sun Yat-Sen University, Shenzhen 518107, China; 2School of Medicine, University of Washington, Seattle, WA 98195, USA; 3Yale School of Public Health, Yale University, New Haven, CT 06520, USA; 4Kirby Institute, University of New South Wales, Sydney, NSW 2052, Australia

**Keywords:** human mpox (monkeypox), mpox virus (MPXV), clinical characteristics, systematic review, meta-analysis, men who have sex with men, HIV

## Abstract

Since May 2022, large numbers of human mpox (previously known as monkeypox) cases have been reported in non-endemic regions. We conducted a systematic review and meta-analysis to elucidate clinical characteristics of the current mpox outbreak. Our systematic review and meta-analysis were undertaken according to PRISMA and MOOSE guidelines. We searched PubMed, EMBASE, and Web of Science for publications between 1 January and 11 November 2022. Random-effects models were used to pool results. Heterogeneity was assessed using *I*^2^. This study is registered with PROSPERO, CRD42022355590. Skin lesions (95.2%, 95% CI [93.3–96.9%]), fever (58.4%, [54.9–61.8%]) and lymphadenopathy (53.0%, [48.7–57.3%]) were the most common symptoms. The most common dermatological manifestations were anogenital lesions (65.7%, [57.8–73.0%]), and the most common lymphadenopathy was inguinal (46.8%, [40.6–53.0%]). There were no differences in symptoms including malaise, fever, headache, and genital, anal, and oropharyngeal lesions according to HIV infection status. Median age of patients varied from 15 to 57.5 years (median, 35 years). The median proportion of men who had sex with men (MSM) was 100.0% (20.6–100.0%). The median proportion of patients who reported recent sexual exposure was 99.2% (14.3–100.0%). The median proportion of PLHIV was 42.2% (0.0–100.0%). Skin lesions, fever, inguinal lymphadenopathy, and anogenital lesions were the most common symptoms of mpox reported in the current outbreak. Existing guidelines should be updated to reflect these clinical manifestations and groups at highest risk of infection, MSM in particular.

## 1. Introduction

Human mpox (previously known as monkeypox), a zoonotic disease caused by the mpox virus (MPXV) [1], was first reported in central Africa in 1970. It occurs primarily in tropical rainforest areas of west and central Africa and was rarely reported elsewhere prior to 2022 [2]. Historically, the clinical presentations of mpox resembled those of smallpox. MPXV is less contagious and causes less severe illnesses compared to smallpox [3]. Since early May 2022, large numbers of mpox cases have been reported in regions where the disease is not endemic, including countries in Europe and North America [4]. Given the continued spread in multiple countries, the World Health Organization (WHO) Director-General declared the global mpox outbreak to be a public health emergency of international concern (PHEIC) on 23 July 2022 [5].

As of 18 November 2022, 80,328 confirmed cases and 53 deaths had been reported in 110 countries, of which 79,355 cases occurred in locations that have not historically reported mpox [6]. MPXV is known to spread through close contact with skin lesions, bodily fluids, contaminated fomites, and large respiratory droplets of infected people or animals. Vertical transmission has also been confirmed in a previous study [1,3]. Recent cases are related predominantly to male–male sexual activity, often reported through sexual health services. While there is no clear evidence that the MPXV can be spread through semen or vaginal fluids, it seems probable that this is contributory [2].

Only one published systematic review has characterized clinical features of mpox in 2022, including 46 studies from 2019 to 2022 [7]. In this review, fever, lymphadenopathy, fatigue, and malaise or asthenia were the most common accompanying symptoms. Lesions on the limbs were the most common dermatological manifestations. With a limited number of studies included, the review conducted a summary of case reports instead of a meta-analysis. Quantitative research is needed for a better review of features of the current multi-country outbreak of mpox that may be distinct from previous outbreaks in endemic areas. A research study suggested that the clinical features and presentations of mpox reported in London differed from previous outbreaks, with lesions on the genital or perianal skin being more common than on the face [8]. Other studies published in 2022 have provided snapshots of mpox clinical symptoms in specific regions and populations, but none have provided an overall description of the clinical features of the current international outbreak. We performed a systematic review and meta-analysis to comprehensively characterize the clinical characteristics of mpox in the current international outbreak. We believe that whether HIV infection leads to more severe mpox clinical symptoms is also a matter of concern. Therefore, we compared mpox symptoms according to HIV infection status. We further sought to identify gaps in identification of clinical characteristics in major mpox prevention and treatment guidelines throughout the world, to better inform efforts to control the epidemic.

## 2. Materials and Methods

### 2.1. Search Strategy and Selection Criteria

Our systematic review and meta-analysis were undertaken according to PRISMA and MOOSE guidelines [9,10]. We searched PubMed, EMBASE, and Web of Science for publications between 1 January and 11 November 2022 by using the following search terms: “monkeypox”, “monkey pox”, “mpox”, “human monkeypox” and “MPX”. Any study design related to the clinical characteristics of patients infected with mpox was considered eligible, including case series, cross-sectional studies, prospective cohort studies, and surveillance reports. Publications in any language were eligible for inclusion. Two investigators (QL and LF) independently screened the literature discovered and assessed each study for inclusion. All duplicates were removed during the study selection process. Disagreements were resolved by consulting the senior investigator (HZ). For our review of public health guidelines, we searched websites of health authorities from different regions, including the WHO, the US Centers for Disease Control and Prevention (CDC), the European CDC, the Nigerian CDC, and the National Health Commission of the People’s Republic of China [6,11,12,13,14]. This study is registered with PROSPERO, CRD42022355590.

### 2.2. Data Extraction and Quality Assessment

A pre-designed form was used to extract study information, including the first author’s name, study design, publication data, patient characteristics (age, proportion of male patients, sexual orientation, hospitalization status, HIV status, sexually transmitted infections, vaccination status, exposure to mpox, clinical symptoms (headache, chill, cough, fatigue/asthenia/malaise, sore throat, fever, myalgia, rectal pain/anal pain/proctitis, rash/exanthema/skin lesions, lymphadenopathy, back pain), the location of skin lesions (oropharyngeal/oral, anogenital/genitals/perianal, upper extremities/arms, lower extremities/legs, face/head, palms and soles/hands or feet, trunk/torso), and the location of lymphadenopathy (inguinal, cervical and axillary). For people living with HIV (PLHIV), we also reviewed antiretroviral treatment, HIV viral load, and CD4^+^ T counts. Six authors extracted all information independently (QL, LF, YL, XW, YL, and YS). We used the Joanna Briggs Institute checklist to assess the risk of bias in all studies included [15,16]. A quality score >5 was considered moderate-high quality as judged independently by two reviewers (QL and YS) [17]. All salient studies were included in the review, regardless of their methodological quality. For guideline review, we collected symptoms of mpox listed in major guidelines identified. Disagreements were resolved through discussion with the senior investigator (HZ).

### 2.3. Data Analysis

#### 2.3.1. Main Analysis

We used random-effect models to calculate the pooled estimated prevalence rates of clinical characteristics and their corresponding 95% confidence intervals (CIs). To reduce the impact of extremely large prevalence (>70%) and extremely small prevalence (<30%), double arcsine conversion was applied to stabilize the variance of specific prevalence rates [18].

The *I*^2^ statistic was used to assess the heterogeneity among studies, with values of 25%, 50%, and 75% representing low, moderate, and high heterogeneity, respectively [19]. We assessed the publication bias by the Egger test where a pooled effect size included three and more studies [20]. The trim-and-fill method was used when the publication bias existed [21]. All computations and data analyses were performed using STATA^®^ 15 (Statistical Software for Data Science; Stata Corp LLC, College Station, TX, USA). We compared symptoms of mpox listed in the above-mentioned major guidelines to the ones identified in our review.

#### 2.3.2. Sensitivity Analysis

We performed a sensitivity analysis to evaluate the effect of eliminating articles with possible overlap. In circumstances where overlapping publications from the same country were identified, only the one with the largest sample size was included. In order to avoid the influence of data of more than 10,000 cases on the results, we excluded these two articles in the second sensitivity analysis.

## 3. Results

We found 6710 publications in our search. After screening for duplicates, 1455 unique publications were retained. An additional 106 publications were further excluded after screening titles and abstracts. We assessed the eligibility of the remaining 1349 studies by reading full-text papers: 1112 did not report clinical features and epidemiological characteristics of mpox; one reported duplicated cases (the one with more detailed information was included), and three reported cases before 6 May 2022 (cases confirmed in 2017–2021), one reported asymptomatic cases. Seventy-seven studies [4,8,22,23,24,25,26,27,28,29,30,31,32,33,34,35,36,37,38,39,40,41,42,43,44,45,46,47,48,49,50,51,52,53,54,55,56,57,58,59,60,61,62,63,64,65,66,67,68,69,70,71,72,73,74,75,76,77,78,79,80,81,82,83,84,85,86,87,88,89,90,91,92,93,94,95,96] with two or more patients were included in our meta-analysis. Seventeen studies [31,34,38,41,45,49,50,55,61,70,83,90,91,92,93,94,95] (four cross-sectional studies that compared symptoms between HIV-infected and HIV-uninfected mpox patients, and 13 case series with information on HIV status) were included in the meta-analysis for comparing symptoms between HIV-infected and HIV-uninfected mpox patients. (Figure 1).

Appendix A summarizes the main characteristics of all included studies. Forty-eight were from Europe, 23 from America, four from Asia, one from 15 countries in Europe, Africa, and America, and one from 16 countries in Europe, Oceania, and America. Among the 77 studies involving 48,622 patients included in our meta-analysis, the number of patients enrolled in each study ranged from two to 26,384. The median age of patients varied from 15 to 57.5 years (median, 35 years; from 62 of 77 studies). The proportion of hospitalized patients ranged from 0.0% to 100.0% (median, 11.1%; 43 studies). The proportion of male patients ranged from 28.6% to 100.0% (median, 100.0%; 75 studies). The proportion of those who had sex with other men (MSM) ranged from 20.6% to 100.0% (median, 100.0%; 49 studies). The proportion of PLHIV ranged from 0.0% to 100.0% (median, 42.2%; 55 studies), among whom 0.0% to 100.0% (median, 94.6%; 22 studies) were on antiretroviral treatment, and CD4^+^ T cell counts (cell/μL) ranged from 50 to 794 (median, 680; 13 studies). The proportion of patients who reported traveling abroad in the month before diagnosis ranged from 0.0% to 100.0% (median, 27.8%; 33 studies). The proportion of patients who attended large gatherings ranged from 0.0% to 75.0% (median, 23.0%; six studies). The proportion of patients who reported recent sexual exposure ranged from 14.3% to 100.0% (median, 99.2%; 42 studies). The proportion of patients who had contact with animals ranged from 0.0% to 76.8% (median, 0.0%; 13 studies). The proportion of patients who had contact with people with similar symptoms ranged from 0.0% to 66.7% (median, 17.0%; 30 studies). The proportion of patients vaccinated against smallpox/mpox ranged from 0.0% to 100.0% (median, 12.1%; 34 studies). Of the 77 studies, 74 were rated as of moderate-high quality, and the remaining three studies were rated as of low quality (Appendix A).

Among the 12 listed symptoms in our meta-analysis (Figure 2 and Figure 3), skin lesions (95.2%, 95%CI [93.3–96.9%)]), fever (58.4%, [54.9–61.8%]) and lymphadenopathy (53.0%, [48.7–57.3%]) were the most commonly reported, followed by fatigue/asthenia/malaise, myalgia, headache, chill, sore throat/odynophagia, rectal pain/anal pain/proctitis, difficulty swallowing/dysphagia, back pain and cough. Forty-three studies reported the proportion of patients who reported rectal pain/anal pain/proctitis, and the pooled estimated prevalence was 18.5% (13.7–27.7%). The most common dermatological manifestations were anogenital lesions (65.7%, [57.8–73.0%]), and the most common lymphadenopathy was inguinal (46.8%, [40.6–53.0%]).

Our meta-analysis showed no differences in malaise, fever, headache, genital lesions, anal lesions, and oropharyngeal lesions between HIV-infected and HIV-uninfected patients (Appendix A). In the HIV-positive group, the proportion of patients on antiretroviral treatment ranged from 0.0% to 100% (median, 99%; nine studies), and CD4^+^ T counts (cell/μL) ranged from 511 to 792 (median, 677; six studies).

Appendix A shows the proportion of STIs among patients with mpox. The proportion of any STI ranged from 0.0% to 100.0% (median, 50.0%; 35 studies). The proportion of patients living with gonorrhea ranged from 0.0% to 50.0% (median, 16.7%; 25 studies). The proportion of patients living with chlamydia ranged from 0.0% to 34.0% (median, 5.3%; 17 studies). The proportion of patients living with syphilis ranged from 0.0% to 100.0% (median, 12.7%; 24 studies). The proportion of patients living with HSV-1 or 2 ranged from 0.0% to 100.0% (median, 0.0%; 14 studies). The proportion of patients living with lymphogranuloma venereum ranged from 0.0% to 1.0% (median, 0.0%; 14 studies). The proportion of patients living with mycoplasma genitalium ranged from 0.0% to 25.0% (median, 4.6%; 12 studies).

Table 1 summarizes differences between mpox symptoms listed in major guidelines and those identified in our review. None of the five guidelines reviewed rectal pain/anal pain/proctitis, four did not mention the specific locations of lymphadenopathy, three did not mention respiratory symptoms, and one did not mention specific locations of skin lesions. The most common skin lesions mentioned by WHO guidelines were facial (in 95% of cases), palms of the hands and soles of the feet (75%), oral mucous membranes (70%), and genitalia (30%).

### 3.1. Publication Bias

Publication bias was found in the meta-analysis for the following symptoms: fatigue/asthenia/malaise, myalgia, chill, headache, axillary lymphadenopathy, anogenital skin lesions, skin lesions on the torso, and oral lesions (Figure 2 and Figure 3, *p* < 0.05). The trim-and-fill method showed a reduced effect estimate for axillary lymphadenopathy, anogenital lesions, and lesions on the torso, while the rest showed no differences (Appendix A).

### 3.2. Sensitivity Analysis

In sensitivity analysis, skin lesions (90.3%, 95%CI [82.6–96.0%], heterogeneity *p* < 0.0001, 29 studies), fever (62.9%, [56.9–68.7%], heterogeneity *p* < 0.019, 27 studies), and lymphadenopathy (55.9%, [50.6–61.1%], heterogeneity *p* < 0.0001, 27 studies) were the most common (Appendix A). The most common type of skin lesions was anogenital lesions (70.6%, [62.0–78.0%], heterogeneity *p* < 0.0001, 27 studies), and the most common kind of lymphadenopathy was inguinal lymphadenopathy (45.4%, [38.1–52.8%], heterogeneity *p* < 0.0001, 34 studies). Chill presented the largest difference of 7.3% after article exclusion, while the rest ranged from 0.0% to 7.1%. In the second sensitivity analysis, chill presented the largest difference of 5.4% after article exclusion, followed by skin lesions on the trunk (3.1%), while the rest ranged from 0.2% to 1.5% (Appendix A).

## 4. Discussion

Our systematic review and meta-analysis found that skin lesions, fever, and lymphadenopathy were the dominant clinical features of the current mpox outbreak in 2022, while fatigue, headache, and myalgia should also be of concern. The most common skin lesions were anogenital, and the most common site of lymphadenopathy was inguinal. Given anecdotes of severe pain with anogenital lesions, we presume that this is an emerging finding that requires future studies to elucidate. We found that MSM, especially those infected with HIV, were the most affected group. There were no differences in symptoms including malaise, fever, headache, genital lesions, anal lesions, and oropharyngeal lesions between HIV-infected and HIV-uninfected patients.

When a given infectious disease emerges or reemerges, accurate identification of clinical characteristics is essential to guiding response efforts. Unlike previous reports from the Democratic Republic of the Congo and Nigeria [8], where rashes appeared first on the face, then across the body, hands, legs, and feet, we found that the most frequently reported cutaneous manifestation was anogenital lesions (65.7%). MPXV can mimic some STIs (e.g., syphilis), leading to misdiagnosis, undertreatment, and continued transmission. Anogenital sites are a key priority when examining patients for potential mpox lesions. The hospitalization rate for this mpox outbreak was about 8%, which is close to that of a current report (13%) at 43 sites in 16 countries in Europe, Oceania, and America [26]. In contrast, a previous systematic review and meta-analysis suggested a hospitalization rate of 35%, which may have been largely biased by its inclusion of studies conducted prior to May 2022 [97]. The MPXV has two clades with different pathogenicity, a milder West African clade and a more virulent Congo Basin clade. Greater representation of the former may explain the low hospitalization rate in the current outbreak [98]. The majority of patients had fever and inguinal lymphadenopathy, though both were seen less often than skin lesions. The lack of specificity of systemic symptoms can pose a challenge to physicians when evaluating patients with possible mpox infections.

Unlike studies published prior to May 2022, we found mpox cases reported in the current outbreak occurred mainly in adult men. In previous outbreaks in Africa, the highest incidence was among children of both sexes who were not vaccinated against smallpox [1]. Compared with an outbreak in the midwestern United States in 2003 (28 years) and endemic cases in Africa reported in the 2010s (21 years), we found that patients in the current outbreak were much older [99,100]. Previous studies have confirmed a trend of increasing median age of mpox cases [2]. Smallpox vaccination was approximately 85% protective against mpox [101]. Routine smallpox vaccination ceased in many regions in the early 1980s following smallpox eradication [99]. Indeed, adults below 40 years are vulnerable to mpox. Although most recent cases occurred after the cessation of routine smallpox vaccination, the older median age in our findings warrants further exploration.

Transmission of MPXV can occur through large respiratory droplets, close or direct contact with skin lesions, and occasionally through contaminated fomites [102]. The likely reason for the historical African outbreaks is that changes in land use brought more people into proximity with virus-carrying animals [1]. Despite previous outbreaks of international imported mpox associated with travelers, secondary human-to-human transmission was rare prior to May 2022 [98]. Genital or oral sex, kissing, and other exchange of bodily fluids are all possible routes of mpox infection, especially in the current outbreak [3]. In the current multi-country outbreak, the proportion of patients who reported recent sexual activity has been relatively high. MSM may be more exposed to large gatherings (such as mass events/parties/sauna) compared to other populations, so larger sexual networks likely accelerate the numbers of exposed and infected persons, as with STIs such as HIV. Reports of primary anogenital lesions support sexual transmission with genitalia serving as primary inoculation sites. Close contact during sex (e.g., skin-to-skin contact and large respiratory droplets) is common, so the primary mode of transmission during sex remains unclear. As persons with classic STI risk factors are a potential risk population for mpox, preventive measures for mpox should be recommended for persons with a history of STIs and/or presenting with traditional STI symptoms. Incorporating mpox screening into routine health management and prioritizing vaccination are priorities for vulnerable populations, including MSM and other sexually active populations with large transmission networks. Given that the preponderance of patients in the current outbreak have been MSM, there are concerns about increased stigma and discrimination against this group, as seen with the HIV epidemic. However, mpox may infect straight men, women, and children alike, so close contact including genital contact and skin-to-skin contact should be the concern, rather than sexuality. Acknowledging that stigma may drive patients away from medical practitioners, attention should be paid to the clinical characteristics of patients to prevent delays in diagnosis in both heterosexual and homosexual populations.

The symptoms of the disease, including malaise, fever, headache, genital lesions, anal lesions, and oropharyngeal lesions, did not differ between patients who were HIV-positive and those who were not. However, a study including 1969 individuals infected with mpox in the United States showed that those infected with HIV were more likely to have rectal pain, rectal bleeding, tenesmus, pus and blood stool, and proctitis. There were still a limited number of studies to be reviewed, so we could not compare more symptoms according to HIV status. Continuous monitoring of outcomes of mpox by HIV infection status is important.

Current guidelines from health authorities have multiple limitations in their description of the symptoms and signs of mpox infection. The sites of skin lesions mentioned in guidelines are consistent with the previous systematic review and meta-analysis, which included studies from 1980–2022 [97]. The frequencies of different skin lesion locations in the previous review were head/neck, 74%; hand palms, 80%; foot soles, 72%; arms/hands 71%; legs/feet, 61%; and mucosae of genitals, 34% [12]. This finding suggests that continuing evidence-gathering is needed to update the guidelines.

Our finding that the most common skin lesions were anogenital lesions differs from those of two previous systematic reviews [7,97]. Benites-Zapata et al. performed a systematic review and meta-analysis with only two of the 12 studies conducted in 2022, so the clinical features shown in that study could not present the features of the current multi-country outbreak. Pourriyahi et al. calculated the prevalence of lesions on the upper and lower limbs together rather than separately, so it may have overlooked the significance of anogenital skin lesions, which are an important characteristic of the current outbreak. Cumulative or average results in this review were calculated with case series and case reports instead of meta-analysis. A total of 80,328 confirmed cases had been reported since 1 January 2022, and the rate of mpox diagnoses slowed after more than 2 months of rapid growth. With enough publications included, we performed a sensitivity analysis to avoid possible overlapping and included 31 additional studies that contributed 48,574 patients to more comprehensively review the current outbreak.

The strength of our systematic review and meta-analysis is their focus on clinical characteristics reported during the current global outbreak. Our review also has several limitations. First, there were few available studies about differences between HIV-infected and HIV-uninfected patients living with mpox, so limited symptoms were analyzed in our meta-analysis. However, our meta-analysis was the first to compare symptoms in mpox-infected patients according to HIV infection status. More research needs to be performed to guide the care and prevention of mpox infection in AIDS patients. Second, laboratory outcomes of this mpox outbreak could not be reviewed because laboratory data were not correlated with clinical data in the literature. Third, the meta-analysis was performed by comparing entire datasets against one another, so we could not analyze data on the level of individual patients.

## 5. Conclusions

Skin lesions, fever, inguinal lymphadenopathy, and anogenital lesions were the dominant clinical features of the current mpox outbreak in 2022. MSM, especially those infected with HIV, are the most affected group. Sexual transmission may be a new transmission route. The ongoing 2022 mpox outbreak is affecting an increasing number of countries. There is an urgent need to update clinical characteristics in mpox guidelines to better inform efforts to control the current global epidemic. Continuous monitoring of outcomes of mpox by HIV infection status is important.

## Figures and Tables

**Figure 1 pathogens-12-00146-f001:**
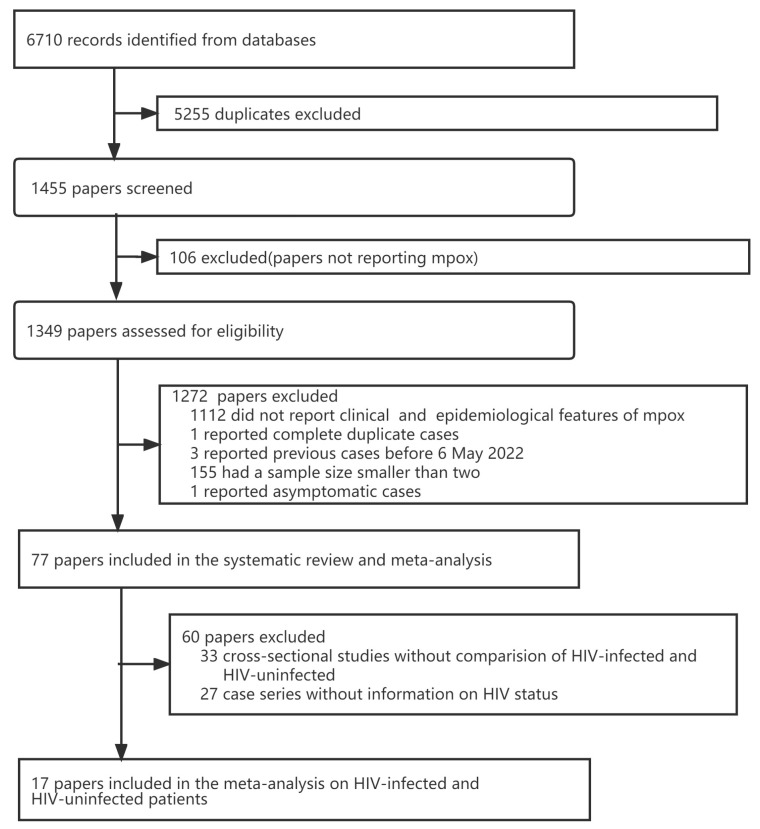
Flowchart of literature search for the clinical characteristics of mpox, May–November 2022.

**Figure 2 pathogens-12-00146-f002:**
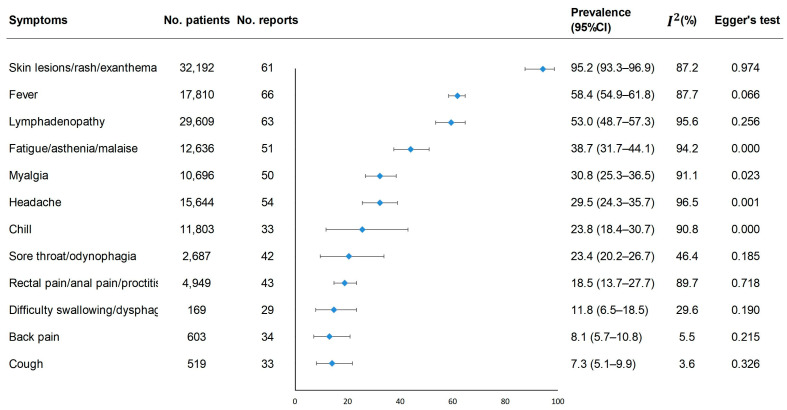
Meta-analysis of the prevalence of clinical symptoms among mpox patients.

**Figure 3 pathogens-12-00146-f003:**
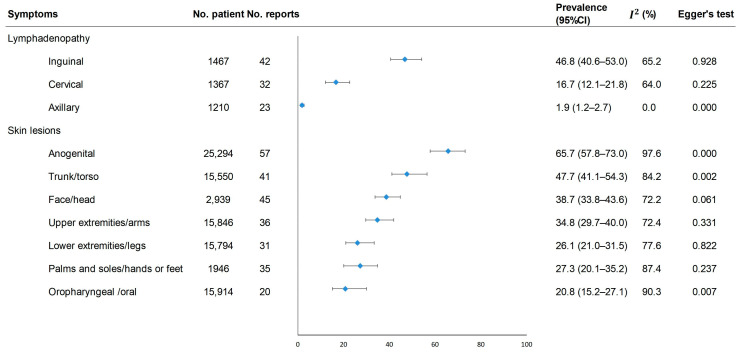
Meta-analysis of the prevalence of rash and lymphadenopathy at different anatomical sites among mpox patients.

**Table 1 pathogens-12-00146-t001:** Discordance between guidelines and our study findings.

Institution *	Missing Information on Clinical Symptoms
World Health Organization	Rectal pain/anal pain/proctitis Respiratory symptoms (cough, sore throat)
European Center for Disease Control and Prevention	Rectal pain/anal pain/proctitis, Information on sites of lymphadenopathy
US Centers for Disease Control and Prevention	Information on sites of lymphadenopathy
Nigeria Center for Disease Control and Prevention	Rectal pain/anal pain/proctitis Respiratory symptoms (cough, sore throat) Information on sites of lymphadenopathy and skin lesions
National Health Commission of the People’s Republic of China	Rectal pain/anal pain/proctitis Respiratory symptoms (cough, sore throat)

* The last updated dates are listed in order: 19 May, 10 June, 18 October, 21 May, and 15 June.

## Data Availability

The data that support the findings of this study are available from the corresponding author, H.Z., upon reasonable request.

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
