# Peer review of "Clinical Characteristics of Human Mpox (Monkeypox) in 2022: A Systematic Review and Meta-Analysis"

_pathogens, 2023, doi:10.3390/pathogens12010146_

Round 1

Reviewer 1 Report

The authors performed a meta-analysis to elucidate the clinical features of the current monkeypox outbreak. The study found that skin lesions, fever, inguinal lymphadenopathy, and anogenital lesions were the most common symptoms of monkeypox reported in the current outbreak. It also emphasized that the population of MSMs deserves attention. Although the clinical symptoms of monkeypox are largely clear, the study still has merits, for example, the authors point out that the most common skin lesions are anogenital, and the most lymphadenopathy is inguinal. Overall, this is an interesting study.

Line 30: Median proportion of men who have sex with men (MSM) was 100.0% (20.6-100.0%). Is the median also 100.0%?

Line 66 - 68: Suggest instead of writing someone suggested, perhaps a research suggested or some studies have shown is a good choice.

monkeypox has been renamed ‘mpox’. Please explain in the corresponding location of the manuscript.

Line 103: CD4+ cell counts: CD4+ T lymphocyte counts. CD4+ T cell counts. +: superscript. T cell or T lymphocyte.

For unknown reasons, the manuscript only shows Figures 1 - 3, Table 1, and Supplementary tables 1 - 4. I can’t see Figures 4 - 8.

Have the authors considered adding funnel plots to visualize publication bias better?

Reviewer 2 Report

The article shows the clinical characteristics of Monkeypox, through a systematic review. The scope of the review is not clear.

Line 2. The title focuses on clinical features of Monkeypox, however the article also compares the clinical features between people living with HIV and those without HIV.

Line 21: The study population is not well defined. At the end of the methodology, it is mentioned that groups of people living with HIV and without HIV were compared.

Line 64. In relation to the previous systematic review. What is new about the current revision? What is the difference with the previous review?

Line 74. The aim of the review is unclear. Are the clinical features of monkeypox? or the comparison between people with and without HIV?

Line 150. 48622 people were studied, but only one study reported 26384. One study had more than half the population analyzed. How does this affect the review?

Line 167. Supplementary table 2 is not mentioned

Line 187. The first 7 supplementary figures are not mentioned.

Line 191. It is inappropriate to start the paragraph by mentioning “Supplementary Table 2“... If it is so important, it should not be in supplementary material.

Line 199. In none of the files or documents of this article, are the supplementary tables. If all the information in the supplementry table is presented in this paragraph, the supplementary table would be unnecessary.

Line 207. It is never mentioned that part of the objective of the work was to detect discrepancies between the clinical characteristics reported by various institutions. Table 1 can be deleted.

Line 223. There is no order in the presentation of the supplementary figures and tables. Supplementary tables are not presented. Are some supplementary figures important? Could they be featured as part of the article?

Line 333. In which part of the article is it shown that people living with HIV are at increased risk of infection? It is reported that 42% of people had HIV, but there are 58% who did not have HIV.

It seems that the aim of the article was to study the clinical characteristics of people living with and without HIV, but it is not clear. The article must focus on an objective, but in the current version it is not clear.

Reviewer 3 Report

I found this manuscript to be well-researched and analyzed and an important contribution to the literature. I appreciated that the authors' described their rationale for why they performed this meta-analysis and how the study under consideration impacted the field at large. From my perspective and level of expertise, this review seems to be methodologically sound.

I have some suggestions for improvement:

- Given the recent guidelines from the WHO, should the authors refer to the virus as "mpox" instead of monkeypox in the manuscript?

- Figures 2 and 3 were a bit hard to read in my version.

- The downloaded materials I viewed did not include any of the referenced Supplementary Tables. All I could see were Supp Figures 4 through 8.

- The Supplementary Tables are referenced out of order. On page 4, the authors reference Supp Table 1 followed by 3 and 4 later in the same paragraph. Supp Table 2 is then referenced later on page 5.

- The Supplementary Figures are also out of numerical order. For example, Supp Fig 8 on page 5 is referenced before 4, 5, 6, or 7, which are mentioned later in the text.

- Should not the Supplementary Figures start with #1 and end with #5? (rather than 4 through 8)

- Please provide a rationale for why HIV-infection status was included in the analysis.

- The authors focused almost exclusively on the current outbreak, which includes cases from all over the world. I think it could be informative to perform an analysis of the clinical presentation based on geographical location.

Reviewer 4 Report

In this meta-analysis, Liu et al., have analyzed clinical characterization of monkeypox, which is endemic in 2022.

They have collected 6710 records from the database. Among these, only 17 publications have analyzed and concluded that skin lesions, fever, inguinal lymphadenopathy, and anogenital lesions are the most common symptoms of monkeypox reported in the current outbreak.    

This meta-analysis has also reported sexual transmission can be a new transmission route, because median proportion of men who have sex with wen is 100%. 

Please include supplementary table 1.

Round 2

Reviewer 2 Report

The authors answered satisfactorily to each of the comments, specially modifyng the supplementary material.